



# Simulated stability of the AMOC during the Last Glacial Maximum under realistic boundary conditions

Frerk Pöppelmeier[1], Jeemijn Scheen[1], Aurich Jeltsch-Thömmes[1], Thomas F. Stocker[1]

[1]Climate and Environmental Physics, Physics Institute and Oeschger Center for Climate Change Research, University of Bern, 3012 Bern, Switzerland

*Correspondence to*: Frerk Pöppelmeier (frerk.poeppelmeier@climate.unibe.ch)

**Abstract.** The response of the Atlantic Meridional Overturning Circulation (AMOC) to freshwater perturbations critically depends on its mean-state. Large swaths of icebergs melting in the North Atlantic during the last deglaciation constituted such perturbations, and thus can provide important constraints on the stability of the AMOC. Yet, the mean AMOC state during the Last Glacial Maximum (LGM), preceding the rapid disintegration of the ice-sheets during the deglaciation, as well as its response to these perturbations remain debated. Here we investigate the evolution of the AMOC responding to freshwater perturbations under improved LGM boundary conditions in the Bern3D intermediate complexity model. Particularly, we consider the effect of an open versus a closed Bering Strait. The vigorous and deep AMOC under these glacial boundary conditions, consistent with previous simulations with different models, reacts more strongly to North Atlantic freshwater forcings than under pre-industrial conditions. This increased sensitivity is mostly related to the closed Bering Strait that cuts off the freshwater escape route through the Arctic into the Pacific, thus facilitating faster accumulation of freshwater in the North Atlantic halting deep water formation. Proxy reconstructions of the LGM AMOC instead indicate a weaker and possibly shallower AMOC than today, in conflict with the particularly strong and deep circulation states coherently simulated with ocean circulation models for the LGM. Simulations with reduced North Atlantic deep water formation, as a consequence of potentially increased continental runoff from ice-sheet melt and imposed changes in the hydrological cycle, more closely resemble the overturning circulation inferred from proxies. These circulation states also show bistable behavior, where the AMOC does not recover after North Atlantic freshwater hosing. However, no AMOC states are found here that either comprise an extreme shoaling or vigorous and concurrent shallow overturning as previously proposed based on paleoceanographic data.



## 1 Introduction

The Atlantic Meridional Overturning Circulation (AMOC) redistributes heat, nutrients, and carbon between the hemispheres and thus constitutes an important tipping element in Earth's climate system (Lenton et al., 2008; Stocker and Wright, 1991). In light of this, painstaking efforts have been devoted to thoroughly understand its sensitivity and response to perturbations that are thought to play a major role in climate variability. Today, the zonally integrated Atlantic circulation is characterized by two overturning cells that are driven by the southward transport of North Atlantic Deep Water (NADW) and northward flowing Antarctic Bottom Water (AABW) occupying abyssal depths. Paleo-reconstructions provide ample evidence that the AMOC experienced extensive reorganizations in the past (Böhm et al., 2015; Broecker and Denton, 1989; McManus et al., 2004; Stocker, 2000). Large AMOC variability is particularly well documented for the transition from the Last Glacial Maximum (LGM, ~20,000 years ago), characterized by ~90 ppm lower atmospheric $CO_2$ concentration (Monnin et al., 2001) and large continental ice-sheets responsible for about 120 m lower sea level (Lambeck et al., 2014), into the current interglacial (Lehman and Keigwin, 1992; McManus et al., 2004). Nonetheless, a large array of uncertainties remains concerning the triggers of these abrupt climate events (Barker et al., 2015) and the overall mean AMOC state during the last glacial that facilitated the rapid sequence of climate changes during the following deglaciation (see Lynch-Stieglitz, 2017 for a review).

Notably, the water mass geometry and circulation strength of the glacial AMOC and Southern Ocean received continuous attention over the past decades in order to improve the understanding of deep ocean carbon storage under the different climatic boundary conditions that prevailed during the LGM. Yet, no consistent framework of the glacial AMOC has emerged to this day (Lynch-Stieglitz, 2017). Early proxy reconstructions suggested a weaker and substantially shallower glacial AMOC (Curry and Oppo, 2005; Duplessy et al., 1988; Fischer et al., 2010; Sarnthein et al., 1994). However, recent investigations provided a first clue that the initial interpretations of these data may have overestimated the extent of shoaling (Gebbie, 2014; Oppo et al., 2018). These findings are corroborated by reconstructions of the AMOC geometry based on Nd isotopes (Du et al., 2020; Howe et al., 2016; Pöppelmeier et al., 2020) providing evidence for little to no change in the overall water mass provenance in the Atlantic between today and the LGM. In addition, reconstructions of the AMOC strength provide equally ambiguous results, either indicating a more vigorous but shallow (Bradtmiller et al., 2014; Lippold et al., 2012) or in contrast strongly weakened deep ocean circulation (Freeman et al., 2016; Skinner et al., 2017).

LGM model simulations in the framework of the third phase of the Paleoclimate Model Intercomparison Project (PMIP3) have not yet helped to reconcile the contrasting AMOC states suggested by proxy reconstructions, as they consistently indicate a stronger and deeper AMOC than during pre-industrial (PI) (Muglia and Schmittner, 2015). This large uncertainty in the mean glacial AMOC state contributes to the lack of understanding of the AMOC's response to freshwater perturbations as they have occurred during the last deglaciation (Heinrich Stadial 1 and Younger Dryas; Broecker, 1992) and may resurface in the future under accelerated warming (Stocker and Schmittner, 1997), or Greenland ice-sheet melting





(Driesschaert et al., 2007). Proxy reconstructions indicate a substantial slowdown of the Atlantic deep circulation during these freshwater discharge events in the past (McManus et al., 2004; Oppo et al., 2015), but quantitative estimates of this weakening remain extremely challenging to obtain due to large proxy uncertainties. In addition, the freshwater fluxes that

drove the slowdown are equally poorly constrained (Clark et al., 2001; Roberts et al., 2014). Taken together, large uncertainties remain in our understanding of the stability of the AMOC.

Here we investigate the impact of critical changes in the boundary conditions between the LGM and PI on the mean AMOC state in the Bern3D Earth system model of intermediate complexity. These changes comprise orbital and radiative forcings, closure of the Bering Strait, changes in the wind stress, and elevated tidal dissipation due to lower sea level

changing shallow-ocean bathymetry. In order to identify processes that affect the stability of these AMOC states we then apply freshwater forcings to the North Atlantic in classical hosing experiments. Finally, we look into the processes that are required to force the model from a mono-stable regime into bistability, with an AMOC that does not recover after freshwater perturbations. This concerns the sensitivity of the hysteresis to model parameters and configurations. The latter may be responsible for transient changes in the hysteresis structure during the transition from the glacial to the Holocene (Stocker

and Marchal, 2000).

## 2 Model description and simulations

### 2.1 Model description

The Bern3D model version 2.0 is an Earth System Model of intermediate complexity with a horizontal resolution of $40 \times 41$ grid cells and 32 logarithmically scaled depth layers (Edwards et al., 1998; Müller et al., 2006; Roth et al., 2014). The

geostrophic-frictional balance ocean model features an isopycnal diffusion scheme and Gent-McWilliams parametrization for eddy-induced transport (Griffies, 1998) and is coupled to a single-layer energy-moisture balance model on the same horizontal grid (Ritz et al., 2011). Wind stress and cloud cover are prescribed from present-day monthly climatologies (ERA40, Kalnay et al., 1996). A prognostic carbon cycle is implemented (Parekh et al., 2008; Tschumi et al., 2011) following the OCMIP-2 protocols (Orr et al., 1999) with updated Schmidt number calculation (Wanninkhof, 2014) and carbon

chemistry (Orr and Epitalon, 2015). We further diagnose the ideal water age through a tracer that is explicitly transported by advection, diffusion, and convection but is set to zero at the surface and else increases with a rate of 1 yr yr$^{-1}$.

For the LGM control simulations the orbital parameters were set to 20 kyr BP (Berger, 1978) and radiative forcing of greenhouse gases was prescribed corresponding to $CO_2$ = 191 ppm, $CH_4$ = 370 ppb, and $N_2O$ = 208 ppb but with dynamic biogeochemistry and $CO_2$ independent of the radiative forcing. Further, the LGM ice-sheet extent and related changes to the

albedo were constrained by reconstructions by Peltier (1994). In order to achieve topologically more realistic glacial boundary conditions we investigate the impact of a closed Bering Strait on the AMOC, which is represented in the model by a single grid cell with a depth of about 40 m and a mean meridional throughflow of 0.5 Sverdrup (1 Sv = $10^6$ m$^3$s$^{-1}$) from the Pacific to the Arctic, which is at the lower end of the observed range of 0.4 to 1.2 Sv (Woodgate et al., 2005). The seasonally





varying wind stress is prescribed in the model based on modern climatologies but was substantially different during the
LGM mostly because the Laurentide and Fennoscandian ice-sheets modulated the northern westerlies (Muglia and
Schmittner, 2015). To alleviate this issue, we calculated LGM anomalies of zonal and meridional wind stresses from five
PMIP3 model outputs of the LGM (CCSM4, CNRM, GISS, MIROC, and MPI; Braconnot et al., 2012) and added the multi-
model mean to the prescribed modern wind stress fields following the approach by Muglia and Schmittner (2015) (Fig.
B2a,b). Finally, due to lower sea level, tidal dissipation and hence diapycnal mixing was substantially increased (1.8-3.0
times) during the LGM (Egbert et al., 2004; Schmittner and Egbert, 2014). To account for this increase in vertical mixing, we
replaced the globally uniform diapycnal diffusivity scheme of the Bern3D model with the output from the UVic model
coupled to the high-resolution tide model OTIS providing 3D diapycnal diffusivity fields for the LGM (Wilmes et al., 2019).
We use the UVic-OTIS simulation with sea levels derived from the ICE-6G database (Fig. B2c). For this simulation the
background diffusivity was kept constant and thus the diapycnal diffusivities can be assumed to represent a conservative
estimate, since effects of remotely dissipated tidal energy are neglected (Wilmes et al., 2019). Table 1 provides an overview
of the model simulations with the according adjustments to the boundary conditions.

Rearrangements of the routing of continental precipitation to the oceans were not performed, since Muglia and
Schmittner (2015) indicated that such changes have little influence on the ocean circulation. Further, average salinities due to
lower sea level were not increased for the LGM simulations, because we do not focus here on globally uniform changes in
salinity and simulations with adjusted salt budget indicate negligible effects on the AMOC (not shown).

**Table 1.** List of simulation setups.

| Simulation | Adjustments |
|---|---|
| PI_CTRL | Pre-industrial boundary conditions. |
| LGM_CTRL | Orbital and radiative forcing of 20 kyr BP. |
| LGM_BS | "LGM_CTRL" + closed Bering Strait. |
| LGM_BS+wind | "LGM_BS" + added wind stress anomaly of LGM from PMIP3 models (Muglia and Schmittner, 2015). |
| LGM_BS+wind+tidal | "LGM_BS+wind" + tidal mixing induced changes in diapycnal diffusivity due to lower sea level (Wilmes et al., 2019). |

**2.2 Freshwater hosing experiments**

In order to test the stability of the AMOC we performed freshwater hosing experiments. For all experiments we applied the
freshwater hosing constantly over 500 years, evenly distributed over the northern North Atlantic between 45° N and 70° N
(Fig. B1). Simulations were hosed with 0.1 to 1.0 Sv of freshwater. The freshwater was not compensated for in the rest of the
ocean, in order to avoid salinity feedbacks elsewhere (Stocker et al., 2007). Moreover, we are interested here in the AMOC



responses of the late glacial/early deglacial to freshwater perturbations (i.e., Heinrich Stadial 1 analogs), which were a net addition of freshwater, hence we consider this approach more realistic.

We also performed a set of experiments with increasing North Pacific to North Atlantic freshwater transfer flux adjustments. This tests the effect of continental ice-sheet runoff and the strength of the Pacific-to-Atlantic freshwater transport via the atmospheric hydrological cycle (Zaucker et al., 1994). By constantly adding 0.02 to 0.12 Sv of freshwater to the North Atlantic and removing the same amount from the North Pacific (i.e., adding salt) we progressively weaken the AMOC.

## 3 Results

### 3.1 Pre-industrial AMOC stability

Under pre-industrial boundary conditions the AMOC strength in the Bern3D model is 17.7 Sv (maximum of the Atlantic overturning stream function below 400 m water depth, Fig. 1a), and the upper circulation cell, defined as positive values in the stream function, reaches down to about 3000 m depth (Fig. 2a). While the AMOC strength is in good agreement with observation-constrained estimates (17.2 Sv; McCarthy et al., 2015), the depth of the upper cell is too shallow compared to observations (Jenkins et al., 2015). During freshwater perturbations the AMOC weakens substantially within 50 years and then evolves somewhat differently depending on the amount of freshwater hosing (Fig. 1a): with 0.1 Sv freshwater hosing the AMOC quickly stabilizes for the duration of the continuous hosing of 500 years, while more than 0.2 Sv freshwater forcing leads to an additional AMOC weakening for another 100 to 200 years. The minimum AMOC strength during hosing

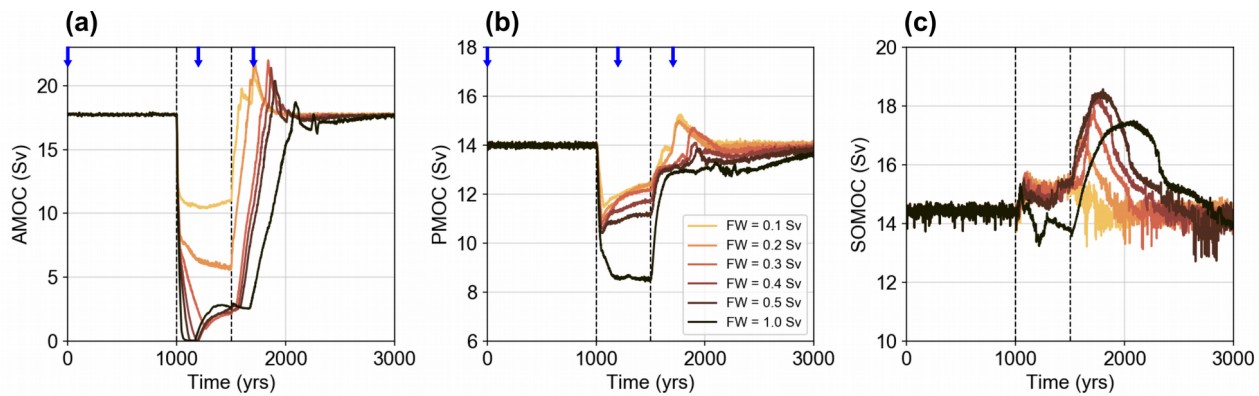

**Figure 1.** Response of circulation strength of (a) AMOC, (b) PMOC, and (c) SOMOC to freshwater hosing experiments for the pre-industrial control. The freshwater (FW) was added for 500 years (start and end are indicated by the vertical dashed lines) to the North Atlantic between 45° N and 70° N (Fig. B1). PMOC and SOMOC circulation cells flow anticlockwise and are hence defined negative by convention (cf. Fig. 2) but are plotted here as positive values for convenience. Note the different y-axes in the panels. Blue arrows in panels a,b mark time steps for which the stream functions are shown in Fig. 2.



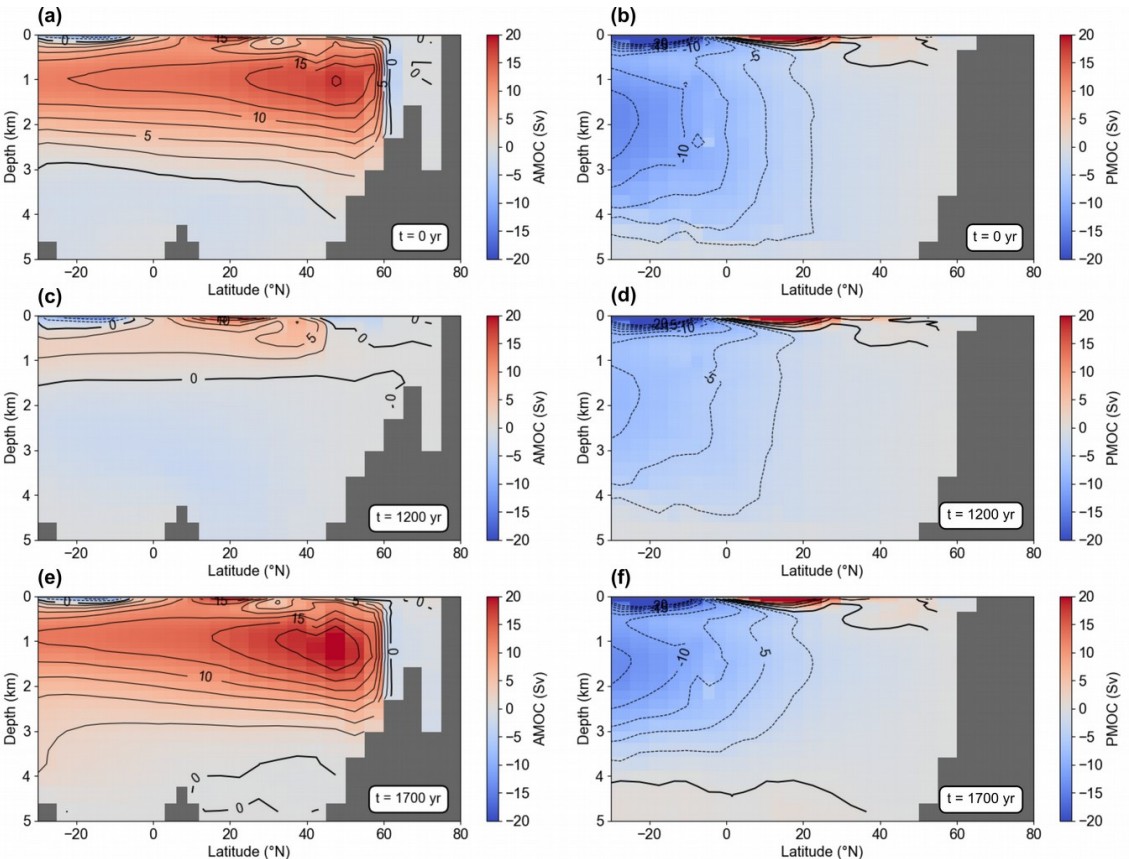

**Figure 2.** Zonally integrated AMOC (left) and PMOC (right) stream functions at different points in time during freshwater hosing of 0.2 Sv as applied in Fig. 1 (orange line) for PI_CTRL. (a,b) Steady-state circulations at time zero of Fig. 1. (c,d) Stream functions at t = 1200 yr, which is 200 years into the freshwater hosing. (e,f) Stream functions at t = 1700 yr, 200 years after the freshwater hosing ended. Positive and negative values correspond to clockwise and anticlockwise circulation, respectively.

does not decrease linearly with the amount of freshwater but instead approaches a collapsed state that slightly recovers during hosing stabilizing at ~2.5 Sv for freshwater perturbations ≥0.3 Sv. For all PI freshwater hosing experiments the AMOC returns to its initial steady-state value within 600 years (time after hosing until the steady-state strength is reached again, i.e., prior to the overshoot), indicating mono-stability of the AMOC. However, this recovery time increases with the amount of freshwater from ~100 years for 0.1 Sv to ~600 years for 1.0 Sv. The Pacific Meridional Overturning Circulation

(PMOC, defined as minimum overturning in the Pacific north of 30° S) is about 14.0 Sv (Fig. 1b), comparable to modern observations of 14.9 Sv (Fig. 2b; McCarthy et al., 2015). The structural response of the PMOC to North Atlantic freshwater hosings is similar to the AMOC, yet with strongly diminished amplitudes in strength reductions. For freshwater amounts ≤0.5 Sv the reduction in PMOC strength is only ~3 Sv while it is about 5.5 Sv for a North Atlantic hosing of 1 Sv, with no indication of a full collapse. Contrastingly, the Southern Ocean Meridional Overturning (SOMOC, defined as the minimum



of the stream function south of 30° S) shows very little change during the 500 years of hosing but increases in strength afterwards by up to 4 Sv for about 500 years before stabilizing again at the steady-state strength.

## 3.2 AMOC under increasingly realistic glacial boundary conditions

Radiative forcing corresponding to lower greenhouse gas concentrations and orbital parameters adjusted to 20 kyr BP with all other boundary conditions kept at PI (i.e., simulation LGM_CTRL) produce an AMOC of 15.8 Sv: about 2 Sv weaker

than PI_CTRL (Fig. 3) and a PMOC of 15.6 Sv slightly stronger than PI_CTRL. This minor AMOC weakening has relatively little impact on the ideal water age distribution in the Atlantic with both PI_CTRL and LGM_CTRL exhibiting deep water ages in the North Atlantic of up to 800 years (Fig. B3; global average of 570 years for LGM_CTRL, Table A1). A more sluggish AMOC is also indicated by nutrient-based reconstructions of the Atlantic circulation during the LGM (e.g., Curry and Oppo, 2005; Lynch-Stieglitz et al., 2007). However, these reconstructions indicate a much older deep ocean as

well as substantial shoaling of the AMOC by up to 1000 m in the North Atlantic, which is not simulated here with virtually no shift in the water depth of the upper circulation cell and only a minor change in ventilation age.

Closing the Bering Strait under the LGM boundary conditions described above (simulation LGM_BS) increases the AMOC strength by 2.3 Sv in comparison to LGM_CTRL by cutting the inflow of relatively fresh water from the North Pacific to the Arctic and subsequently to the deep water formation zones of the North Atlantic. As such, the closure of the

Bering Strait produces an AMOC of 18.1 Sv and thus slightly stronger than PI_CTRL with major increases below 1000 m water depth (Fig. 3b and d), at the same time also increasing the PMOC to about 16.2 Sv. This strengthening also deepens the AMOC by about 500 m in comparison to LGM_CTRL, contrasting the notion of a shoaled glacial AMOC suggested by nutrient-based proxy reconstructions. The combination of both the strengthening and deepening leads to better ventilation of the deep North Atlantic compared to LGM_CTRL yielding ideal water ages between 400 and 500 years as well as a global

average ideal water age of 555 years (Fig. B3).

In a next step, adding the PMIP3 LGM wind stress anomalies to the PI wind field (simulation LGM_BS+wind) further increases the circulation strength of the AMOC and PMOC by an additional 1.4 Sv and 1.7 Sv yielding a total of 19.5 Sv and 17.9 Sv, respectively. This also further deepens the AMOC by about 1000 m, now reaching down to the ocean bottom at 5000 m in the northern North Atlantic. As a consequence, ideal water ages are <200 years in the deep North

Atlantic and 400-500 years in the deep South Atlantic, while the global average is about 30 years younger than in LGM_CTRL. The stronger and southward shifted northern hemispheric westerlies of the LGM, caused by the large continental ice-sheets, increase the strength of the subpolar and subtropical gyres. This in turn enhances the northward salt flux to the northern North Atlantic subsequently intensifying deep water formation (Muglia and Schmittner, 2015). Finally, we also consider the effect of lower glacial sea level shifting tidal dissipation from the continental shelfs to the deep ocean.

Here this is parametrized by replacing the diapycnal diffusivity of the Bern3D model with the ones derived from the UVic ocean model coupled to the OTIS tide model (Wilmes et al., 2019) (i.e., simulation LGM_BS+wind+tidal), producing an



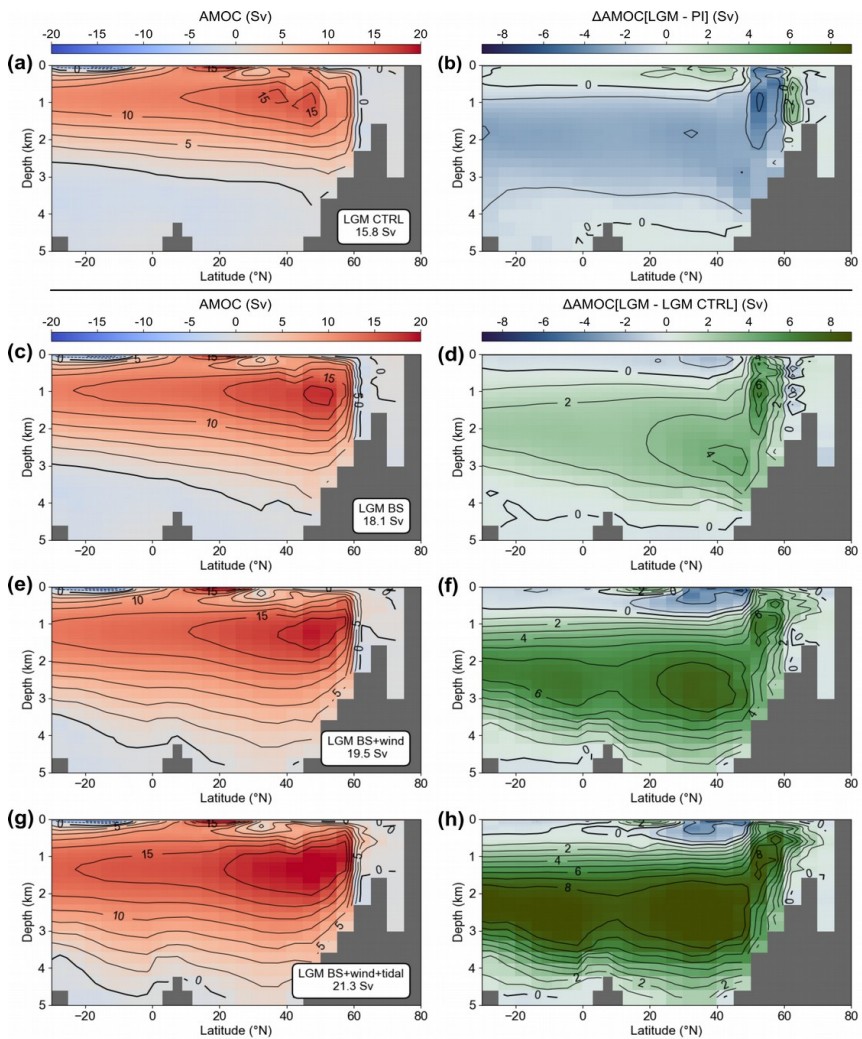

**Figure 3.** (a) Zonally integrated overturning stream function of LGM_CTRL (AMOC = 15.8 Sv) and (b) difference to the PI_CTRL depicted in Fig. 2a. (c,d) LGM with closed Bering Strait (18.1 Sv) and difference to LGM_CTRL (panel a). (e,f) LGM with closed Bering Strait and LGM wind stress (19.5 Sv). (g,h) LGM with closed Bering Strait, LGM wind stress, and increased diapycnal diffusivity due to lower sea level (21.3 Sv). See Sec. 3 and Table 1 for details on simulations.

even stronger and slightly deeper AMOC (21.3 Sv) and PMOC (18.0 Sv) than LGM_BS+wind. These effects can be related to the elevated tidal dissipation increasing the downward mixing of northern sourced water as well as promoting mixing
between NADW and AABW (Wilmes et al., 2019). This also slightly increases deep Atlantic (and Pacific) ventilation yielding a global mean ideal water age of 490 years thus about 80 years younger than LGM_CTRL. In sum, these arguably more realistic LGM boundary conditions produce an AMOC that is about 20 % stronger and more than 1500 m deeper than PI_CTRL. These results are hence in relatively good agreement with the PMIP3 LGM simulations yielding an average increase in AMOC of 41 ± 26 % and an average deepening of 665 ± 550 m (Muglia and Schmittner, 2015).





### 3.3 LGM freshwater hosing experiments and AMOC hysteresis


The various LGM configurations are expected to have different stability properties with respect to freshwater perturbations. In order to illustrate this, we apply freshwater discharge (hosing) to the North Atlantic, which leads to different responses of the AMOC in the different LGM model configurations (Fig. 4). First, we compare the different LGM configurations for a freshwater hosing of 0.2 Sv for 500 years (Fig. 4a). In simulations with an open Bering Strait (PI_CTRL and LGM_CTRL)
the hosing reduces the AMOC strength by about 12 Sv, while a reduction of ≥15 Sv is observed for the LGM simulations with a closed Bering Strait. The physical origin of this difference lies in the freshwater balance in the North Atlantic. An open Bering Strait acts as a buffer that provides relatively salty Pacific water (relative to the hosing perturbation) to the North Atlantic during hosing when the Arctic/North Atlantic sea surface salinity is decreased, essentially diminishing the impact of the freshwater discharge. When the Bering Strait is closed, this buffer is absent and freshwater accumulates more
easily in the North Atlantic during hosing. This increases upper ocean stratification and thus prevents deep water formation more effectively. However, large freshwater perturbations >0.3 Sv overwhelm this buffering process and a collapsed circulation state with a residual circulation of about 2 Sv is approached regardless of the configuration (Fig. 4b). Recovery of the circulation also varies substantially between the configurations with fast recoveries within ~100 years for PI_CTRL, LGM_BS+wind, and LGM_BS+wind+tidal while the steady-state circulation is reached again only after 300 and 500 years
for LGM_BS and LGM_CTRL, respectively. These recoveries are characterized by AMOC overshoots that also greatly vary in magnitude from 2 Sv for LGM_CTRL to 9 Sv for LGM_BS+wind+tidal. Overall, the magnitude of the overshoots correlates well with the steady-state AMOC strength of the respective configuration.

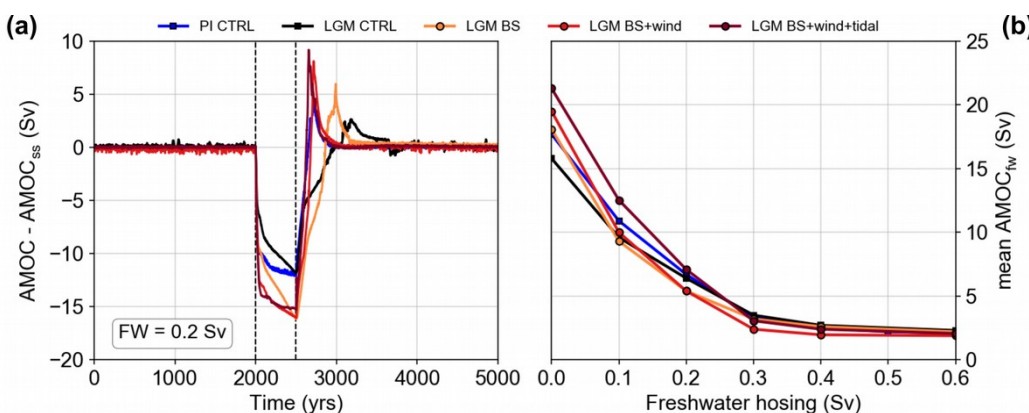

**Figure 4.** (a) AMOC responses to freshwater perturbation of 0.2 Sv for 500 years in the North Atlantic for the different simulations listed in Table 1. Vertical dashed lines mark the start and end of hosing. To compare the AMOC anomalies the steady-state AMOC strength (AMOC$_{ss}$) has been subtracted from the AMOC signal. (b) 500-year average of AMOC strength during the freshwater perturbation (AMOC$_{fw}$) versus amount of freshwater hosing.



These different responses to freshwater are also reflected in the structure of the AMOC hysteresis, which was assessed by applying a freshwater forcing to the North Atlantic between 45° N and 70° N linearly increasing at a rate of 0.1 Sv/kyr (Fig. 5). After reaching a maximum freshwater flux of 1 Sv the forcing was gradually decreased to zero at the same rate. This small rate allows the AMOC to adjust to the freshwater flux and reach a quasi-equilibrium. Both PI_CTRL and LGM_CTRL simulations with an open Bering Strait exhibit relatively little hysteresis, due to a smaller North Atlantic salt anomaly. A

collapsed state is reached for both these simulations for a hosing of about 0.23 Sv and recovery starts below 0.15 Sv. In fully coupled atmosphere-ocean general circulation models (AOGCMs) this negative feedback of the Bering Strait emerges from the difference in sea level between the North Pacific and Arctic that reverses during freshwater hosing, i.e., export from the Arctic to the North Pacific during hosing in contrast to an import during steady-state (Hu et al., 2012). The Bern3D model has a rigid-lid ocean and the Bering Strait throughflow is always into the Arctic. Yet, both effects (freshwater export to the

Pacific in AOGCMs and import of relatively saltier water from the Pacific in the Bern3D model) reduce the North Atlantic salt anomaly during freshwater hosing reducing the AMOC hysteresis. With a closed Bering Strait in simulation LGM_BS+wind+tidal the AMOC responds to a freshwater forcing with a two-step reduction on top of the continuous decrease. The first accelerated reduction occurs at 0.08 Sv hosing, followed by a second steep decline in AMOC strength at 0.25 Sv of freshwater discharge leading towards a full collapse of the circulation. After that, the AMOC stays in the

collapsed state for more than 16 kyrs and only recovers when the hosing decreases below 0.08 Sv, followed by an overshoot reaching a circulation of more than 25 Sv until it returns to its initial steady-state. These structures are reminiscent of the hysteresis behavior of simple box models and theoretical considerations, which are driven by Stommel's salt advection feedback (Fig. 5a; Rahmstorf et al., 2005; Stocker and Wright, 1991; Stommel, 1961). The stronger hysteresis of LGM_BS+wind+tidal can be traced back to the increased freshwater accumulation in the North Atlantic due to the closed

Bering Strait (cf. Hu et al., 2012). Once collapsed, the circulation is unable to efficiently export the freshwater anomaly elsewhere. Thus, the restart of the AMOC is delayed, since the only North Atlantic source of salt is via the subpolar gyre. As such, this larger AMOC hysteresis supports the notion that the closure of the Bering Strait played a central role for the abrupt climate transitions during the last glacial such as Dansgaard-Oeschger events (Hu et al., 2012).

**3.4 Effect of AMOC strength on freshwater response**

Proxy reconstructions of the LGM AMOC strength are strongly diverging, with some studies indicating a more vigorous but shallow (Bradtmiller et al., 2014; Lippold et al., 2012) and others a more sluggish circulation (Curry and Oppo, 2005; Evans and Hall, 2006; Lynch-Stieglitz et al., 2007). Yet, the model simulations presented here as well as PMIP3 results indicate a stronger and deeper Atlantic circulation in conflict with these proxy reconstructions. In order to also explore the responses of potentially weak glacial AMOC states to freshwater perturbations, we continuously apply North Pacific-to-North Atlantic

freshwater transfer fluxes with all other boundary conditions as in LGM_BS+wind+tidal, which we consider the configuration closest to the actual LGM boundary conditions. This yields reduced steady-state AMOC strengths from 20.4 to





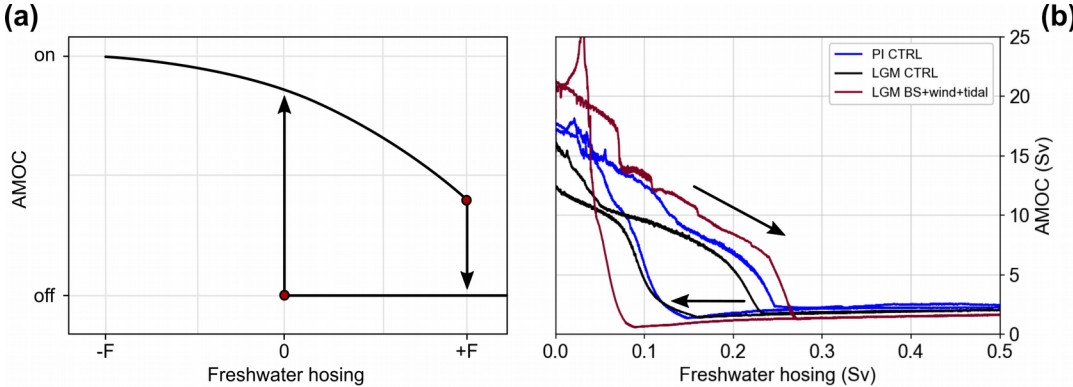

**Figure 5.** (a) Schematic hysteresis structure of the AMOC with bifurcation points (red circles, Stommel, 1961). (b) Hysteresis diagram for PI control (blue), LGM control (black), and most realistic LGM boundary conditions (red, LGM_BS+wind+tidal). Here the freshwater was linearly increased up to 1.0 Sv over a period of 10 kyrs and then decreased back to zero over 10 kyrs.

12.3 Sv (freshwater transfer between 0.02 and 0.12 Sv; Fig. 6). These freshwater adjustments, mimicking increased runoff from the continental ice-sheets and changes in evaporation and precipitation, lead to substantial shoaling of the AMOC due to increased upper ocean stratification (Fig. B4). However, the shoaling does not extent beyond the PI_CTRL AMOC depth

even for the weakest AMOC state produced by a freshwater transfer flux of 0.12 Sv. Further, despite the decrease of AMOC strength of 9 Sv (and 2.8 Sv for PMOC) between these simulations, the impact on ventilation ages is relatively minor with an increase by only ~30 years yielding a global mean of 520 years for the simulation with the weakest AMOC compared to LGM_BS+wind+tidal, which is still about 80 years younger than PI_CTRL.

        Hosing these weakened AMOC states with 0.2 Sv freshwater for 500 years decreases the minimum AMOC strength

during the perturbation with increasing freshwater adjustments (Fig. 6). Simulations with a steady-state AMOC larger than 15 Sv recover to their initial state, but the recovery time increases from ~200 years for the smallest freshwater adjustment to about 1000 years for an adjustment of 0.08 Sv after the perturbation. This indicates that in the LGM_BS+wind+tidal model configuration the AMOC is more sensitive to freshwater perturbations with decreasing overturning strength corroborating the findings by Goes et al. (2019). Indeed, for an even larger Pacific-to-Atlantic freshwater adjustment leading to an AMOC

strength of <15 Sv (Adj = 0.10 Sv), the freshwater hosing of 0.2 Sv causes a total collapse of the circulation, which does not recover after the freshwater hosing. This demonstrates bistability under these conditions and hence a completely different response to perturbations.

        Overall, this bistable behavior is present in all LGM model configurations (Fig. 7), but the transition from mono- to bistability (red regimes) is more abrupt for LGM_BS+wind and particularly LGM_BS+wind+tidal. In both these LGM

configurations the AMOC either recovers relatively quickly or collapses completely. In contrast, in LGM_CTRL and LGM_BS the AMOC requires up to 2800 years to recover without fully collapsing for weak AMOC steady-states and large freshwater forcings. This highlights that the LGM wind stress anomalies and elevated vertical mixing are the driving processes leading to this abrupt behavior between either fast recovery or full collapse.



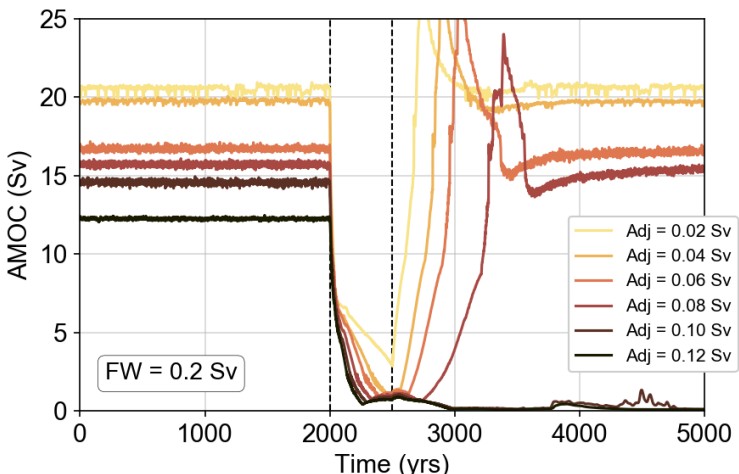

**Figure 6.** Sensitivity of weakened AMOC states of LGM_BS+wind+tidal boundary conditions to freshwater perturbations of 0.2 Sv over 500 years. In order to reduce the AMOC strength Pacific-to-Atlantic freshwater transfer fluxes of 0.02 to 0.12 Sv were applied to the North Atlantic, which were compensated for in the North Pacific (Fig. B1).

### 3.5 Response in atmospheric $CO_2$ concentration

In order to diagnose the response in atmospheric $CO_2$ concentrations to these different circulation states, we make use of the prognostic carbon cycle implemented in the Bern3D model (Parekh et al., 2008; Tschumi et al., 2011). The transition from PI_CTRL to LGM_CTRL boundary conditions lowers the dynamically simulated atmospheric $CO_2$ concentration from 278 ppm to 252 ppm (Table 2; note that the LGM radiative greenhouse gas forcing is prescribed to be equivalent to $CO_2$ = 191 ppm, $CH_4$ = 370 ppb, and $N_2O$ = 208 ppb and is independent of the dynamically simulated atmospheric concentrations). This

26 ppm decrease is less than a third of the observed reduction of about 90 ppm (Monnin et al., 2001) indicating that these simplified changes in boundary conditions lack important processes that were responsible for the lower glacial atmospheric $CO_2$ concentration. Yet, simulation LGM_BS+wind+tidal with arguably more realistic physical boundary conditions exhibits even higher $pCO_2$ of 258 ppm, mainly due to the considerably more vigorous ocean circulation and the associated increase in deep ocean ventilation. This difference of 6 ppm $pCO_2$ between LGM_CTRL and LGM_BS+wind+tidal is relatively small

considering the increase in AMOC strength of 5.5 Sv (and 4.4 Sv increase in PMOC). Similarly, the simulations with weakened AMOC states (Fig. 6) also only lower $pCO_2$ again down to 255 ppm for the weakest AMOC of 12.3 Sv, indicating a weak sensitivity of $pCO_2$ to the AMOC strength in the Bern3D model, which is also reflected in the fairly small changes in ideal water age between these simulations. This is partly related to the substantially smaller changes in PMOC strength, representing more than twice the ocean volume of the AMOC, that decreases by <3 Sv between the LGM_BS+wind+tidal

simulations with 0 and 0.12 Sv Pacific-to-Atlantic freshwater transfer fluxes. Further, Southern Ocean overturning increases concurrently with decreasing AMOC strength, thus counteracting the weaker ventilation due to reduced NADW formation. As such, these simulations indicate that the Atlantic overturning is not the main driver for lower glacial $pCO_2$ in the Bern3D model, thus contradicting a number of studies suggesting a dominant role of Atlantic deep water reorganizations for lower LGM $CO_2$ concentrations (e.g., Sigman et al., 2010). Instead, we surmise that the full range of the LGM-PI $CO_2$ difference





can only be explained in the Bern3D model by applying a combination of changes in both physical and biogeochemical forcings as suggested by Menviel et al. (2012) and Jeltsch-Thömmes et al. (2019). For example, changes in the remineralization depth of organic material in the ocean, which is currently parameterized according to Martin et al. (1987), exert a large impact on atmospheric $CO_2$ (Jeltsch-Thömmes et al., 2019; Kwon et al., 2009; Menviel et al., 2012; Roth et al., 2014). Another important process that is thought to have contributed significantly to the enhanced glacial oceanic carbon

sequestration is the greatly elevated glacial dust flux fertilizing regions starved of bioavailable iron, particularly in the Southern Ocean (Khatiwala et al., 2019; Lambert et al., 2015). Further, ocean-sediment interactions, such as $CaCO_3$ compensation and imbalances in the weathering-burial cycle of carbon, alkalinity, and nutrients, cannot be neglected on glacial-interglacial timescales as they exert a strong control on carbon cycle changes (Roth et al., 2014, Jeltsch-Thömmes et al., 2019).

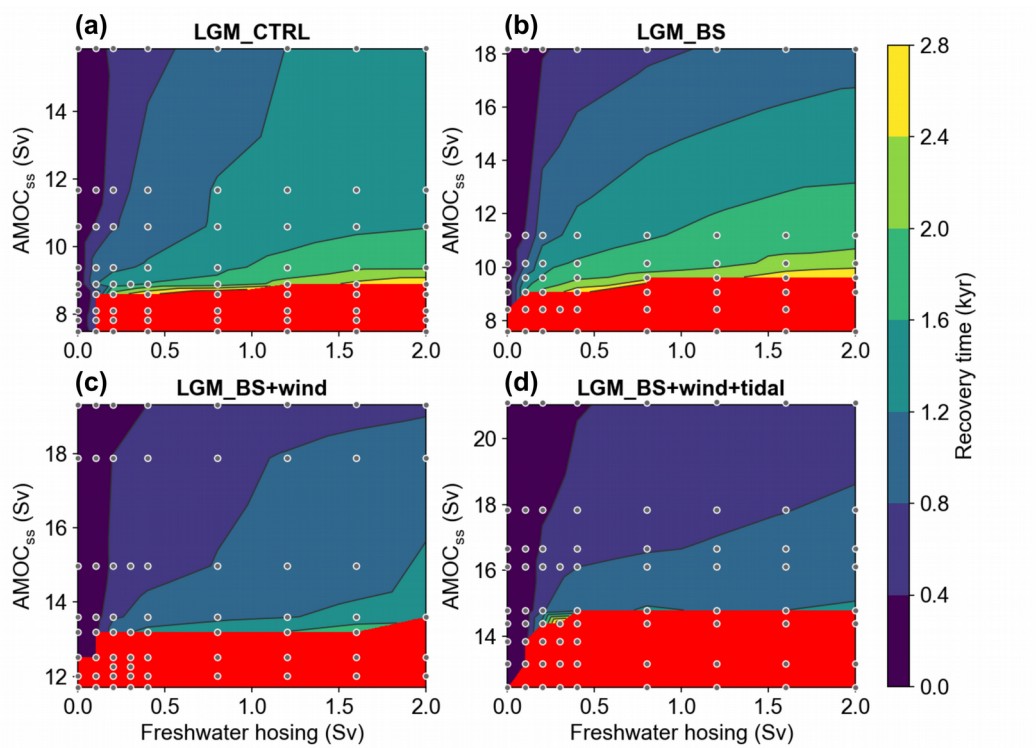

**Figure 7.** Dependency of AMOC recovery time on steady-state circulation strength ($AMOC_{ss}$) and freshwater hosing amplitude. Model configurations of (a) LGM_CTRL, (b) LGM_BS, (c) LGM_BS+wind, and (d) LGM_BS+wind+tidal. The simulations are indicated by gray dots. The AMOC was weakened by applying a freshwater transfer flux from the North Pacific to the North Atlantic (same as for Fig. 6). Freshwater hosing applied to the North Atlantic (45° N–70° N) lasted for 500 years. The recovery time was calculated as the time after the hosing until the steady-state AMOC strength was reached again. Red areas mark simulations where the AMOC remained in the collapsed state after the freshwater perturbation with no indication of recovery. Note the different scales on the y-axes.




**Table 2.** Atlantic, Pacific, and Southern Ocean Meridional Overturning rates (AMOC, PMOC, and SOMOC, respectively), and atmospheric $CO_2$ concentrations of all simulations.

| Run | AMOC (Sv) | PMOC (Sv) | SOMOC (Sv) | $pCO_2$ (ppm) |
|---|---|---|---|---|
| PI_CTRL | 17.7 | 14.0 | 14.4 | 278 |
| LGM_CTRL | 15.8 | 15.6 | 24.2 | 253 |
| LGM_BS | 18.1 | 16.4 | 22.4 | 247 |
| LGM_BS+wind | 19.5 | 17.6 | 21.1 | 252 |
| LGM_BS+wind+tidal | 21.3 | 17.9 | 23.1 | 259 |
| Adj = 0.02 Sv | 20.6 | 17.7 | 23.3 | 258 |
| Adj = 0.04 Sv | 19.8 | 17.3 | 23.6 | 258 |
| Adj = 0.06 Sv | 16.7 | 16.7 | 24.1 | 257 |
| Adj = 0.08 Sv | 15.7 | 16.3 | 24.6 | 257 |
| Adj = 0.10 Sv | 14.6 | 15.8 | 25.2 | 256 |
| Adj = 0.12 Sv | 12.3 | 15.2 | 26.0 | 255 |

### 3.6 Transient opening of the Bering Strait

Finally, we provide a first glance of the effect of transient changes in the state configuration of the Bern3D model. After massive continental ice-sheet melt during the last deglaciation, sea level rose rapidly first reconnecting the North Pacific with the Arctic during the Younger Dryas cold event (Pico et al., 2020). Here we simulate the opening of the Bering Strait in a transient simulation under glacial boundary conditions (i.e., LGM_BS+wind+tidal; Fig. 8). Results of this simulation should be taken with caution and can only describe first-order consequences of the late deglacial Bering Strait opening, since

the boundary conditions used here have limited validity for that time. More realistic transient boundary conditions of the late deglaciation are desirable, but the complex interactions between the effects of the retreating continental ice-sheets with ocean circulation (freshwater fluxes, changes in the wind field, differences in tidal dissipation due to sea level rise) and the consequent transient changes are difficult to constrain and beyond the scope of this study.

The opening of the Bering Strait rapidly decreases the mean Arctic salinity (integrated over all depths) due to the

inflow of relatively fresh Pacific surface water (Fig. 8a and B6). Subsequently, the salinity slightly recovers after ~100 years reaching a new steady-state after ~800 years that is about 0.05 psu less saline. This negative salt anomaly also propagates to the North Atlantic somewhat disrupting deep water formation and reducing the AMOC by about 1 Sv (Fig. 8b). The evolution of the AMOC is similar to the mean Arctic salinity, yet the recovery to the new steady-state only takes ~400 years. This new steady-state after the opening of the Bering Strait exhibits an AMOC that is stronger than that of the steady-state

with an open Bering Strait (Fig. B6). Further, the AMOC variability increases significantly after the Bering Strait opening



from ± 0.06 Sv (1σ from 800-1000 yrs) to ± 0.25 Sv (1800-2000 yrs). Overall, the opening has relatively little impact on the sea surface salinity of the North Atlantic (Fig. B6) and thus on the AMOC strength in the Bern3D model under LGM boundary conditions. Consequently, this simulation suggests that the considerable climate and AMOC reorganizations found by proxy reconstructions for the late deglaciation (Denton et al., 2010; McManus et al., 2004) require further forcing

mechanisms such as freshwater fluxes from the decaying Laurentide and Fennoscandian ice-sheets (Condron and Winsor, 2012; Keigwin et al., 2018; Renssen et al., 2015).

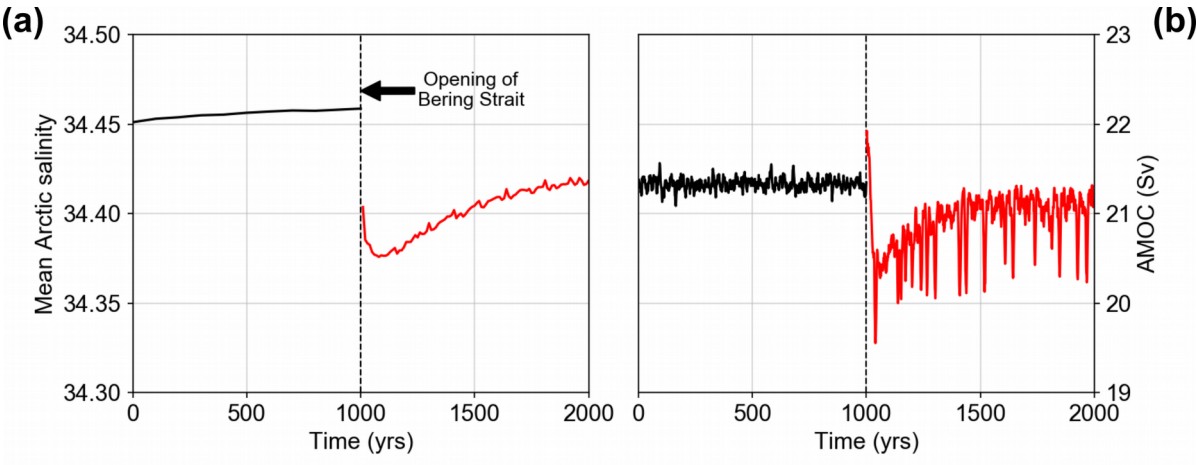

**Figure 8.** (a) Evolution of mean Arctic salinity (integrated over all water depths) and (b) AMOC strength with opening of the Bering Strait at year 1000. The mean meridional advection through the Bering Strait is 0.5 Sv.

## 4 Discussion and Conclusion

Despite decades of research, large uncertainties remain on the overall geometry and strength of the AMOC during the LGM (Lynch-Stieglitz, 2017). In conflict with proxy reconstructions, PMIP3 model simulations consistently indicate a stronger

and deeper AMOC under LGM boundary conditions than PI (Muglia and Schmittner, 2015). Here we confirm these results of the PMIP3 simulations with the Bern3D model, indicating that three major differences in the boundary conditions between LGM and PI even overcompensate for the AMOC weakening associated with the lower glacial temperatures such that the resulting LGM state exhibits a stronger AMOC than PI. Our simulations emphasize that the closure of the Bering Strait, wind stress anomalies, and changes in tidal dissipation all have substantial impact on the ocean circulation here

leading to a total increase in AMOC strength of ~5.5 Sv and as such need to be considered for paleoclimate model simulations. When investigating abrupt climate transitions during the last glacial, it is particularly important to consider the interaction between the North Pacific and Arctic. Closing the Bering Strait increases the sensitivity of the AMOC to freshwater perturbation by preventing the negative feedback of salt/freshwater export through the Bering Strait (Hu et al.,





2012). Hence, freshwater can accumulate in the North Atlantic more easily, and upper ocean stratification rapidly increases
and suppresses deep water formation.

A counteracting effect to the AMOC strengthening due to the Bering Strait closure, increased wind stress, and elevated tidal dissipation during the LGM is the potentially increased runoff from the continental ice-sheets surrounding the North Atlantic or changes in the hydrological cycle that are not explicitly implemented here. Our simulations indicate that such an additional freshwater flux could have substantially weakened and shoaled the AMOC. Yet, even for large freshwater
relocations (>0.1 Sv) the AMOC does not substantially shoal beyond the PI AMOC depth in the Bern3D model. The strongly increased vertical mixing due to higher tidal dissipation, often neglected in climate simulations, and elevated North Atlantic wind stress counterbalance the increased stratification due to the freshwater flux (Fig. B4; Wilmes et al., 2019). Therefore, the simulations presented here suggest that the LGM AMOC did not shoal to the extent previously inferred from nutrient-based proxies (Curry and Oppo, 2005; Lynch-Stieglitz et al., 2007). Instead, relatively high nutrient concentrations in the
deep North Atlantic, interpreted as an indicator for southern sourced water, are presumably related to changes in the carbon cycle, i.e., increased remineralization of organic matter and changes in the air-sea gas exchange (Gebbie, 2014; Howe et al., 2016). Furthermore, in our model an AMOC weakening is always accompanied by an AMOC shoaling and vice versa, and no circulation is found here that produces a strong and shallow AMOC as initially suggested by $^{231}$Pa/$^{230}$Th reconstruction (Bradtmiller et al., 2014; Lippold et al., 2012).

While the work presented here suggests that some proposed LGM circulation regimes seem to be difficult to realize, further constraints from proxy reconstructions are required to reduce uncertainties and thus better determine the LGM AMOC state. With the present study, we have established a consistent model framework with easily tunable circulation strength that will facilitate comprehensive model-data intercomparisons with geochemical (Pa/Th, Nd isotopes) and nutrient-based ($\delta^{13}$C, $\Delta^{14}$C) circulation proxies implemented in the Bern3D model in the future.



## Appendix A: Model initialization and ideal ages

### A1 Model initialization

The Bern3D model was spun up over 35,000 years to a pre-industrial (1765 CE) equilibrium comprising greenhouse gas concentrations of $CO_2$ = 278 ppm, $CH_4$ = 722 ppb, and $N_2O$ = 273 ppb. For the LGM_CTRL simulation the model was further spun up over 10,000 years continuing from PI_CTRL to orbital parameters adjusted to 20,000 years BP (Berger, 1978) and greenhouse gas concentrations of $CO_2$ = 191 ppm, $CH_4$ = 370 ppb, and $N_2O$ = 208 ppb. After the PI spin up, atmospheric $CO_2$ concentrations could freely evolve in response to e.g., the ventilation state of the ocean and corresponding $CO_2$ sequestration or outgassing, but radiative forcing remained prescribed as before. For all simulations with a closed Bering Strait (LGM_BS, LGM_BS+wind, and LGM_BS+wind+tidal) this same procedure was performed with the sole difference of the changed bathymetry at the Bering Strait, respectively different wind stress fields and/or diapycnal diffusivities. The freshwater experiments were then started from the corresponding spin-ups.

### A2 Ideal ages for different simulations

We diagnosed the mean ideal ages for the global ocean as well as for the different sub-basins below 2 km water depth (Table A1). These ages measure the true model ocean ventilation and thus cannot be directly compared to radiocarbon ventilation ages that exhibit a preformed surface age due to partly not fully equilibrating at the ocean-atmosphere interface, sea-ice prohibiting air-sea gas exchange, and carbon cycle related changes.

**Table A1.** Mean ideal water ages for the global ocean and the different sub-basins.

| Run | Global mean (yr) | Atlantic: >2 km (yr) | Pacific: >2 km (yr) | Indic: >2 km (yr) | Southern Ocean: >2 km (yr) |
|---|---|---|---|---|---|
| PI_CTRL | 614 | 479 | 1003 | 758 | 529 |
| LGM_CTRL | 570 | 483 | 943 | 635 | 403 |
| LGM_BS | 555 | 338 | 939 | 594 | 396 |
| LGM_BS+wind | 526 | 211 | 908 | 567 | 396 |
| LGM_BS+wind+tidal | 488 | 178 | 870 | 542 | 374 |
| Adj = 0.02 Sv | 490 | 186 | 869 | 549 | 373 |
| Adj = 0.04 Sv | 491 | 200 | 866 | 557 | 371 |
| Adj = 0.06 Sv | 513 | 297 | 887 | 603 | 396 |
| Adj = 0.08 Sv | 514 | 331 | 886 | 609 | 394 |
| Adj = 0.10 Sv | 514 | 369 | 883 | 612 | 390 |
| Adj = 0.12 Sv | 518 | 457 | 879 | 625 | 388 |



**Appendix B: Figures**

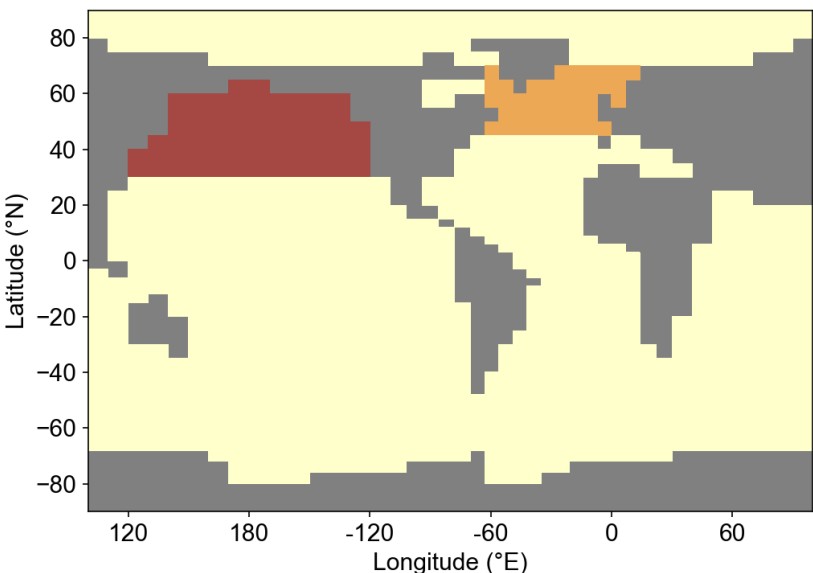

**Figure B1.** Map depicting regions where freshwater/saltwater perturbations are applied. For the hosing experiments the freshwater was evenly applied to the North Atlantic (orange). In order to achieve weakened LGM AMOC states small freshwater fluxes were added to the North Atlantic (orange) and compensated for in the North Pacific (red) by adding the equivalent salt flux to that region.







**Figure B2.** Additional changes in LGM boundary conditions. (a,b) Multi model mean wind stress anomalies of five PMIP3 models (CCSM4, CNRM, GISS, MIROC, and MPI) following Muglia and Schmittner (2015). The large positive anomaly south of Australia is a regridding artifact due to the lack of Tasmania in the Bern3D model. (c) Zonally averaged diapycnal diffusivities of the Atlantic, which are regridded on the Bern3D grid from the UVic climate model coupled to the OTIS tide model with sea level derived from ICE-6G (Wilmes et al., 2019).








**Figure B3.** Zonally averaged ideal water ages of the Atlantic for the different model configurations listed in Table 1. Ideal ages are set to zero at the surface and increase in the ocean interior with a rate of 1 yr yr$^{-1}$.



**Figure B4.** (left) Zonally integrated Atlantic stream functions with all LGM forcing changes (simulation LGM_BS+wind+tidal) but freshwater adjustment of 0.04 Sv to 0.12 Sv between the North Pacific and North Atlantic (cf. Fig. 6). The regions of the freshwater flux and according salt compensation are marked in Fig. B1. (right) Zonally averaged ideal water age in the Atlantic. Ideal ages are set to zero at the surface and increase in the ocean interior with a rate of 1 yr yr$^{-1}$.

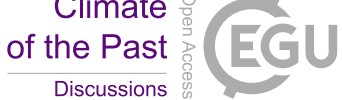

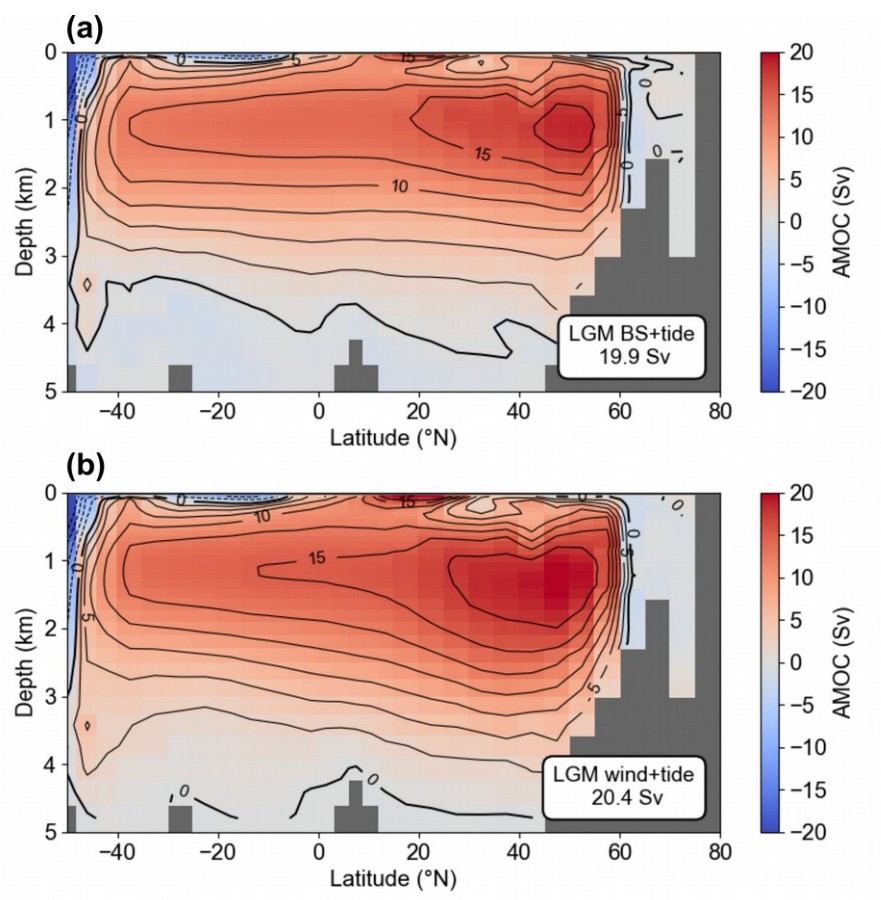

**Figure B5.** Atlantic stream functions for LGM boundary conditions with (a) closed Bering Strait and consideration of increased tidal dissipation after Wilmes et al. (2019) and (b) added LGM wind stress anomalies from PMIP3 models (Muglia and Schmittner, 2015) and increased tidal dissipation.





**Figure B6.** Annual mean sea surface temperature (SST), sea surface salinity (SSS), and fractional sea-ice north of 50° N under LGM boundary conditions. (top) Closed Bering Strait (simulation LGM_BS+wind+tidal), (middle) new steady-state after transiently opening the Bering Strait but all other boundary conditions as in top panel, and (bottom) difference between open and closed Bering Strait.


*Code availability.* Simulation outputs used for this study are available upon request from the corresponding author (frerk.poeppelmeier@climate.unibe.ch).

*Author contributions.* FP and TFS designed the study. FP developed and performed the model simulations with the help of AJT and JS. FP wrote the initial manuscript with contributions from all co-authors.



***Competing interests.*** The authors declare that they have no conflict of interest.

***Acknowledgments.*** Calculations were performed on UBELIX, the HPC cluster at the University of Bern. This is TiPES contribution #45. FP and TFS acknowledge financial support from the European Union's Horizon 2020 research and innovation program under grant agreement No 820970 (project TiPES). TFS and JS received financial support from the Swiss National Science Foundation (SNSF grant 200020-172745). AJT acknowledges funding from SNSF grant 200020-
172476 and from the European Union's Horizon 2020 research and innovation program under grant agreement No 820989 (project COMFORT, Our common future ocean in the Earth system – quantifying coupled cycles of carbon, oxygen, and nutrients for determining and achieving safe operating spaces with respect to tipping points). The work reflects only the authors' view; the European Commission and their executive agency are not responsible for any use that may be made of the information the work contain. Finally, we thank Fortunat Joos for fruitful discussions.

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
