# Peer review of "Simulated stability of the AMOC during the Last Glacial Maximum under realistic boundary conditions"

_Climate of the Past, 2020_

## Referee Comment (RC1) · Anonymous Referee #1 · 8 Nov 2020

Poppelmeier et al. study the AMOC response to glacial boundary conditions and meltwater input into the North Atlantic with a 3D model. In agreement with previous modelling studies, they find that appropriate glacial boundary conditions lead to a stronger and deeper AMOC at the LGM than during the pre-industrial control. A weaker and shallower LGM AMOC can be obtained by enhanced freshwater flux to the North Atlantic. They further assess the stability of the AMOC to different amounts of North Atlantic meltwater input under pre-industrial and glacial boundary conditions. It is a well written and interesting study, but I suggest to perform an additional experiment with changes in diapycnal diffusivity, more details need to be given about the experimental set up, and the study needs to take into account/discuss previous work done

on the subject. Please find below a few suggestions.

1) Introduction and discussion: The authors briefly describe the LGM AMOC state as inferred from paleo-proxy records (L. 43-50) as well as the LGM AMOC state as simulated by PMIP3 models (L. 51-53), however no mention is made of the work done by combining modelling work and paleo-data as in for example Hesse et al., 2011 (Paleoceanography), Gebbie 2014 (Paleoceanography, even though a rapid mention to this work is given later in the manuscript), Menviel et al., 2016 (Paleoceanography) and Menviel et al., 2020 (Paleoceanography) .

Maybe more importantly, L. 332-336, the experiments presented here cannot provide conclusions on the state of the oceanic circulation at the LGM. There is nothing in the manuscript that can justify the statement on L. 333 about the AMOC depth, and there is no argument either for the statement on L. 336, since the present simulations performed with the Bern3D are not compared to paleo-proxy records and the carbon cycle response to the changes is not studied here.

2) Changes in diapycnal diffusivity The impact of changes in diapycnal diffusivity on the AMOC strength and stability are studied. This is interesting but for consistency, it would have been better to compare the LGM-tidal to the CTL-tidal-PI. tidal -PI being the pre-industrial diapycnal diffusivity values as estimated from the UVic-OTIS. Indeed, it is stated that in CTL diapycnal diffusivity is globally uniform (what is the value of the globally uniform diapycnal diffusivity used in the Bern3D?). Applying a varying diapycnal diffusivity in the CTL might also impact the AMOC. The impact on oceanic properties of the varying diapycnal diffusivity should be mentioned.

3) North Pacific to North Atlantic freshwater flux The reasoning behind increasing the North Pacific to North Atlantic freshwater flux by up to 0.12 Sv is unclear. It is stated that this test the impact of increased runoff from glacial ice-sheets. So, effectively this is equivalent to the freshwater hosing, which is fine in principle. The problem is when the additional freshwater is added, that it can become confusing: L. 244-247: from a

[Figure]

LGM adjusted state of 0.1 Sv, 0.2 Sv is added into the North Atlantic, does that mean that effectively 0.3 Sv are added at that time? The model is forced (for how many years?) with an "adjusted Pac to Alt. fwf flux" of 0.1 Sv, after which the "adjusted flux" is stopped and 0.2 Sv of meltwater are added into the North Atlantic (hosing) for 500 years. At the end of the 500 years is the Atlantic flux back at 0 Sv or 0.1 Sv. This is particularly important to clarify for Figure 7, which is a bit confusing, as in each column at least the initial freshwater flux (or even the total flux) are different for each dot.

4) Impact of AMOC changes on atmospheric CO2 concentration The results are interesting, but they are not discussed at all: there is no explanation as to why the concentration of atm. CO2 changes, and how. In addition, there is no mention of the extensive literature on the topic of the impact of AMOC changes on atm. CO2 (e.g. Schmittner et al.,2008, Menviel et al., 2014, Yu et al., 2016).

Figure 6: The y axis should be adjusted so that none of the lines are cut.

---

## Referee Comment (RC2) · Anonymous Referee #2 · 23 Nov 2020

In this paper, Pöppelmeier et al. present model simulations of the Atlantic Meridional Overturning Circulation (AMOC) at the Last Glacial Maximum (LGM) under different boundary conditions and model configurations. They use the Earth System Model of Intermediate Complexity BERN3D to explore the sensitivity of the LGM AMOC state to changes in wind stress, freshwater fluxes, Bering Strait and diapycnal diffusivity. Consistently with previous model results they find a stronger AMOC under realistic LGM boundary conditions. The paper is well written and the results are mostly clearly presented. The paper is of general interest for readers of Climate of the Past and fits the scope of the journal. I therefore recommend publication of the paper after the comments below (mostly minor) have been addressed.

1) Model configuration and setup would benefit from a more detailed description. I would for instance suggest to move the Appendix A1 to the main text, otherwise in the methods section there is no information on how the model is initialized etc. How is the carbon cycle initialized? What carbon cycle setup is used? Is it only ocean carbon cycle interacting with a one-box atmosphere? Are ocean sediments included? What about land carbon cycle and weathering? Is there any kind of interactive vegetation? Does $CO_2$ affect any land properties?

2) Does topography have an effect in the model (besides of the effect on wind stress)? Would it be sensitive to different LGM ice sheet reconstructions?

3) Some general information on the simulated LGM climate would be useful to get an idea of how the model performs. What are e.g. changes in global temperature, ocean temperature, sea ice area in both Hemispheres?

4) The paper section describing the response in atmospheric $CO_2$ concentration is poorly described and is missing a discussion of the ample available literature on the relation between AMOC and $CO_2$ and more generally on the LGM carbon cycle, largely based on EMICs, with some of the studies even using the same model (BERN3D) (e.g. (Ganopolski and Brovkin, 2017; Kemppinen et al., 2019; Menviel et al., 2017)). Additionally, as already mentioned above, very little information on the carbon cycle setup is given in the paper.

5) There is no comparison of the different AMOC states with available reconstructions of isotopes in the ocean that would allow to make some statement of the likelihood of the different states. This is a pity considering that BERN3D includes several isotopes that could be used to constrain plausible LGM AMOC configurations, although the authors mention that this is in their future plans. Maybe the whole section dealing with simulated atmospheric $CO_2$ concentration would fit better in one of those future papers...?

Line 8: were there really more icebergs during the deglaciation than at LGM?

Lines 26-27: I don't understand the logic behind this sentence. How is the fact that the AMOC is important for the Earth system related to it being (possibly) a tipping point?

Line 87: is the 0.5 Sv Bering Strait throughflow prescribed or computed by the model?

Line 305: where does this variability originate from in the model. Is it just noise? I guess there should be no interannual variability in a model like BERN3D...

Fig. 5b: red and black lines are very hard to distinguish, at least for color blind people.

References:

Ganopolski, A. and Brovkin, V.: Simulation of climate, ice sheets and CO2 evolution during the last four glacial cycles with an Earth system model of intermediate complexity, Clim. Past, 13(12), 1695–1716, doi:10.5194/cp-13-1695-2017, 2017.

Kemppinen, K. M. S., Holden, P. B., Edwards, N. R., Ridgwell, A. and Friend, A. D.: Coupled climate-carbon cycle simulation of the Last Glacial Maximum atmospheric CO2 decrease using a large ensemble of modern plausible parameter sets, Clim. Past, 15(3), 1039–1062, doi:10.5194/cp-15-1039-2019, 2019.

Menviel, L., Yu, J., Joos, F., Mouchet, A., Meissner, K. J. and England, M. H.: Poorly ventilated deep ocean at the Last Glacial Maximum inferred from carbon isotopes: A data-model comparison study, Paleoceanography, 32(1), 2–17, doi:10.1002/2016PA003024, 2017.

---

## Author Comment (AC1) · 18 Dec 2020

**Response to reviewer comments: Simulated stability of the AMOC during the Last Glacial Maximum under realistic boundary conditions (cp-2020-135)**

We are grateful to both reviewers for the evaluations of our work and the particularly helpful and constructive reviews. In the following, we tried to answer all questions raised and incorporated the requested changes in the revised manuscript. We believe this has produced a manuscript with more robust conclusions and more thorough discussion of our results.

The original reviewer comments are in black and our responses are colored blue. Line references correspond to the original manuscript. Rephrased or added paragraphs are marked in italics.

**Reviewer#1:**

Poppelmeier et al. study the AMOC response to glacial boundary conditions and meltwater input into the North Atlantic with a 3D model. In agreement with previous modelling studies, they find that appropriate glacial boundary conditions lead to a stronger and deeper AMOC at the LGM than during the pre-industrial control. A weaker and shallower LGM AMOC can be obtained by enhanced freshwater flux to the North Atlantic. They further assess the stability of the AMOC to different amounts of North Atlantic meltwater input under pre-industrial and glacial boundary conditions. It is a well written and interesting study, but I suggest to perform an additional experiment with changes in diapycnal diffusivity, more details need to be given about the subject. Please find below a few suggestions.

Reply#1: We thank the reviewer for the helpful comments and suggestions. As mentioned in the detailed replies below, we endeavored to amend the manuscript accordingly, which also includes the suggested additional experiment with variable diapycnal diffusivity under pre-industrial boundary conditions.

1) Introduction and discussion: The authors briefly describe the LGM AMOC state as inferred from paleo-proxy records (L. 43-50) as well as the LGM AMOC state as simulated by PMIP3 models (L. 51-53), however no mention is made of the work done by combining modelling work and paleo-data as in for example Hesse et al., 2011 (Paleoceanography), Gebbie 2014 (Paleoceanography, even though a rapid mention to this work is given later in the manuscript), Menviel et al., 2016 (Paleoceanography) and Menviel et al., 2020 (Paleoceanography). Maybe more importantly, L. 332-336, the experiments presented here cannot provide conclusions on the state of the oceanic circulation at the LGM. There is nothing in the manuscript that can justify the statement on L. 333 about the AMOC depth, and there is no argument either for the statement on L. 336, since the present simulations performed with the Bern3D are not compared to paleo-proxy records and the carbon cycle response to the changes is not studied here.

Reply#2: We thank the reviewer for the suggestion to incorporate a number of studies discussing combined model-data approaches in the introduction. We agree that mentioning these lines of work as well, provides a more complete overview of the subject of this study. Therefore, we included an

additional paragraph in the introduction, which discusses the studies by Hesse et al. (2011) and Menviel et al. (2017; 2020).

We further agree that the statement in L. 334-336 cannot be fully substantiated by our simulations in this manuscript, since  $\delta^{13}$ C was not included in the simulations conducted here. We therefore removed this sentence in the revised manuscript. However, we want to note that our simulations do indeed provide important physical constraints on the AMOC state during the LGM. Only considering realistic boundary conditions, we cannot force the model into a state as was initially interpreted from stable carbon isotope reconstructions, i.e., an extreme shoaling of North Atlantic Deep Water. This is also in line with revised interpretations of the updated  $\delta^{13}$ C data in follow-up studies by partly the same authors (Curry and Oppo, 2005 versus Oppo et al., 2018; Keigwin and Swift, 2017). Since we already mention in L. 332 that this conclusion is only suggested by our simulations and hence might be model specific, we prefer to keep this statement in its current form.

2) Changes in diapycnal diffusivity The impact of changes in diapycnal diffusivity on the AMOC strength and stability are studied. This is interesting but for consistency, it would have been better to compare the LGM-tidal to the CTL-tidal-PI. tidal -PI being the pre-industrial diapycnal diffusivity values as estimated from the UVic-OTIS. Indeed, it is stated that in CTL diapycnal diffusivity is globally uniform (what is the value of the globally uniform diapycnal diffusivity used in the Bern3D?). Applying a varying diapycnal diffusivity in the CTL might also impact the AMOC. The impact on oceanic properties of the varying diapycnal diffusivity should be mentioned.

Reply#3: As suggested by the reviewer, we conducted an additional model simulation with the diapycnal diffusivities of the UVic-OTIS model replacing the globally uniform value of  $2 \times 10^{-5}$  m2/s (now also mentioned in the main text) for pre-industrial boundary conditions (Fig. R1). The overall differences are small, with the AMOC being slightly weaker by ~0.7 Sv and slightly shallower in the simulation with the diapycnal diffusivities from the UVic-OTIS model. We now added Fig. R1 to the appendix and included the results of this simulation in the main text: "*In order to verify that these changes are not related to the different parametrizations, we also performed an additional experiment for the pre-industrial where we replaced the globally uniform diapycnal diffusivity of the Bern3D model with the 3D UVic-OTIS model results present-day tides (Wilmes et al., 2019), while keeping all other parameters as in PI\_CTRL (Fig. B4). With the replaced diapycnal diffusivities the AMOC strength is slightly reduced by ~0.7 Sv and only to a small extent shallower than in PI\_CTRL and hence the observed changes in LGM\_BS+wind+tidal cannot be attributed to the different parametrizations."*

**Figure R1:** (top) Stream function for model run with the globally uniform diapycnal diffusivity of PI\_CTRL replaced with the 3D diapycnal diffusivity field of the UVic-OTIS model with present day tides (Wilmes et al., 2019). (bottom) Difference between top panel and simulation PI\_CTRL. The AMOC strength is ~0.7 Sv weaker and slightly shallower in the simulation with the 3D diapycnal diffusivity field of the UVic-OTIS model.

3) North Pacific to North Atlantic freshwater flux. The reasoning behind increasing the North Pacific to North Atlantic freshwater flux by up to 0.12 Sv is unclear. It is stated that this test the impact of increased runoff from glacial ice-sheets. So, effectively this is equivalent to the freshwater hosing, which is fine in principle. The problem is when the additional freshwater is added, that it can become confusing: L. 244-247: from a LGM adjusted state of 0.1 Sv, 0.2 Sv is added into the North Atlantic, does that mean that effectively 0.3 Sv are added at that time? The model is forced (for how many years?) with an "adjusted Pac to Alt. fwf flux" of 0.1 Sv, after which the "adjusted flux" is stopped and 0.2 Sv of meltwater are added into the North Atlantic (hosing) for 500 years. At the end of the 500 years is the Atlantic flux back at 0 Sv or 0.1 Sv. This is particularly important to clarify for Figure 7, which is a bit confusing, as in each column at least the initial freshwater flux (or even the total flux) are different for each dot.

Reply#4: We apologize for any confusion caused by our wording on the North Pacific-to-North Atlantic freshwater transfer flux. We mention in L. 232 that this represents changes in the hydrological cycle not explicitly simulated by the simplified energy-moisture balance model as well as background continental runoff from meltwater. The main differences between the transfer flux and the freshwater hosing is that the former is compensated for by adding salt to the North Pacific. It thus represents an additional tunable branch of the atmospheric hydrological cycle that influences the Pacific-to-Atlantic salinity contrast and is solely for the purpose of sensitivity experiments. This flux is applied constantly while the hosing flux is a true perturbation that is not compensated for and

only applied over 500 years as an additional forcing. All simulations with North Pacific-to-North Atlantic freshwater transfer fluxes were first run into equilibrium over 5000 years before the experiment with North Atlantic freshwater hosing was started. Thus, the transfer flux is basically used here as a tuning parameter for the AMOC strength in our simulations. We tried to explain this now in the main text as clear as possible.

4) Impact of AMOC changes on atmospheric CO2 concentration The results are interesting, but they are not discussed at all: there is no explanation as to why the concentration of atm. CO2 changes, and how. In addition, there is no mention of the extensive literature on the topic of the impact of AMOC changes on atm. CO2 (e.g. Schmittner et al., 2008, Menviel et al., 2014, Yu et al., 2016).

Reply#5: Both reviewers mention that the section on the  $pCO_2$  responses requires additional discussion including isotopes and model-data intercomparisons. We agree with this, but feel that such an investigation would qualify as a stand-alone study and is beyond the scope of the present manuscript. We therefore followed the suggestion of reviewer#2 to remove this section from the manuscript. Please see also reply#11 for more details.

Figure 6: The y axis should be adjusted so that none of the lines are cut. Reply#6: Adjusted.

**Reviewer#2**

In this paper, Pöppelmeier et al. present model simulations of the Atlantic Meridional Overturning Circulation (AMOC) at the Last Glacial Maximum (LGM) under different boundary conditions and model configurations. They use the Earth System Model of Intermediate Complexity BERN3D to explore the sensitivity of the LGM AMOC state to changes in wind stress, freshwater fluxes, Bering Strait and diapycnal diffusivity. Consistently with previous model results they find a stronger AMOC under realistic LGM boundary conditions. The paper is well written and the results are mostly clearly presented. The paper is of general interest for readers of Climate of the Past and fits the scope of the journal. I therefore recommend publication of the paper after the comments below (mostly minor) have been addressed.

Reply#7: We thank the reviewer for the positive evaluation of our work and are grateful for the detailed comments that helped to significantly improve the manuscript.

1) Model configuration and setup would benefit from a more detailed description. I would for instance suggest to move the Appendix A1 to the main text, otherwise in the methods section there is no information on how the model is initialized etc. How is the carbon cycle initialized? What carbon cycle setup is used? Is it only ocean carbon cycle interacting with a one-box atmosphere? Are ocean sediments included? What about land carbon cycle and weathering? Is there any kind of interactive vegetation? Does CO2 affect any land properties?

Reply#8: We agree with the reviewer on this point and we therefore moved the detailed description of the model initialization from the Appendix to the model description in Sect. 2 of the main text.

Based on the comments from reviewer#1 and the suggestion of point 5 of reviewer#2, we removed the discussion on the carbon cycle response (previous Sect. 3.5). Please see also reply#11 regarding this. Accordingly, it is obsolete to provide further information on the initialization of the carbon cycle.

2) Does topography have an effect in the model (besides of the effect on wind stress)? Would it be sensitive to different LGM ice sheet reconstructions?

Reply#9: The Bern3D model is currently not coupled to an ice-sheet model and therefore changes in ice-sheet reconstructions only have a very limited impact on the model (only on the corresponding land-albedo), besides obvious changes such as closures of ocean gateways e.g., the Bering Strait that we investigate in detail. Wilmes et al. (2019) showed that the tidal mixing and hence the diapycnal diffusivity is fairly sensitive to the glacial ice-sheet extent. However, since tidal mixing is not dynamically simulated in the Bern3D model, we can unfortunately not test this.

3) Some general information on the simulated LGM climate would be useful to get an idea of how the model performs. What are e.g. changes in global temperature, ocean temperature, sea ice area in both Hemispheres?

Reply#10: We added the information on global mean surface temperature, mean ocean temperature, and sea ice extent of both hemispheres to Table 2.

4) The paper section describing the response in atmospheric CO2 concentration is poorly described and is missing a discussion of the ample available literature on the relation between AMOC and

CO2 and more generally on the LGM carbon cycle, largely based on EMICs, with some of the studies even using the same model (BERN3D) (e.g. (Ganopolski and Brovkin, 2017; Kemppinen et al., 2019; Menviel et al., 2017)). Additionally, as already mentioned above, very little information on the carbon cycle setup is given in the paper.

5) There is no comparison of the different AMOC states with available reconstructions of isotopes in the ocean that would allow to make some statement of the likelihood of the different states. This is a pity considering that BERN3D includes several isotopes that could be used to constrain plausible LGM AMOC configurations, although the authors mention that this is in their future plans. Maybe the whole section dealing with simulated atmospheric CO2 concentration would fit better in one of those future papers...?

Reply#11: We agree with the reviewer that the discussion on the carbon cycle changes would greatly benefit from model-data comparisons of carbon isotope reconstructions. However, investigating carbon isotopes would require us to redo all simulations. Further, additional processes important for carbon cycling, such as weathering fluxes and sediment-bottom water carbon exchange, would need to be considered (e.g., Jeltsch-Thömmes et al., 2019). We feel that this is not only far beyond the scope of this study but would also not fit well in the research context of the present manuscript where we focus on the physical changes in ocean circulation related to increasingly realistic boundary conditions. We therefore followed the suggestion of the reviewer and removed the discussion section on the atmospheric  $pCO_2$  response from the manuscript. Yet, we want to note that this study will form the basis for future investigations where we plan to study geochemical and carbon isotope proxies in a combined approach in order to elucidate carbon cycle changes since the LGM in a comprehensive way, as also mentioned in the last sentence of Sect. 4.

Line 8: were there really more icebergs during the deglaciation than at LGM?

Reply#12: Based on ice-rafted debris counts from the North Atlantic (e.g., Hodell et al., 2017), icebergs were indeed much more common during the last deglaciation, in particular during Heinrich Event 1 and the Younger Dryas, than during the LGM.

Lines 26-27: I don't understand the logic behind this sentence. How is the fact that the AMOC is important for the Earth system related to it being (possibly) a tipping point?

Reply#13: We agree that this sentence was unclear and deleted the word 'thus'. It now reads: "*The Atlantic Meridional Overturning Circulation (AMOC) redistributes heat, nutrients, and carbon between the hemispheres and constitutes an important tipping element in Earth's climate system.*"

Line 87: is the 0.5 Sv Bering Strait throughflow prescribed or computed by the model? Reply#14: The Bering Strait throughflow is computed by the model. We now clarified this in the main text.

Line 305: where does this variability originate from in the model. Is it just noise? I guess there should be no interannual variability in a model like BERN3D...

Reply#15: It is correct that the model variability is not a consequence of interannual variability modes such as ENSO or NAO, but instead originates mainly from internal, localized salt oscillations. We feel that it already becomes clear in the model description of Sect. 2 that such

interannual variability modes cannot be represented in the Bern3D model with its relatively coarse grid resolution.

Fig. 5b: red and black lines are very hard to distinguish, at least for color blind people.

Reply#16: We are sorry for any inconvenience caused by our choice of colors. We have now adopted an improved color scheme for Figs. 4 and 5 that should provide a better contrast also for dichromatic color blindness.

**References**

- Curry, W. B., & Oppo, D. W. (2005). Glacial water mass geometry and the distribution of  $\delta^{13}$ C of  $\Sigma$ CO2 in the western Atlantic Ocean. Paleoceanography, 20(1), 1–12.
- Hesse, T., Butzin, M., Bickert, T., & Lohmann, G. (2011). A model-data comparison of  $\delta^{13}$ C in the glacial Atlantic Ocean. Paleoceanography, 26(3), 1–16.
- Hodell, D. A., Nicholl, J. A., Bontognali, T. R. R., Danino, S., Dorador, J., Dowdeswell, J. A., Einsle, J.,Kuhlmann, H., Martrat, B., Mleneck-Vautravers, M. J., Rodríguez-Tovar, F. J., & Röhl, U. (2017).Anatomy of Heinrich Layer 1 and its role in the last deglaciation. Paleoceanography, 32(3), 284–303.
- Jeltsch-Thömmes, A., Battaglia, G., Cartapanis, O., Jaccard, S. L., & Joos, F. (2019). Low terrestrial carbon storage at the Last Glacial Maximum: Constraints from multi-proxy data. Climate of the Past, 15(2), 849–879.
- Keigwin, L. D., & Swift, S. A. (2017). Carbon isotope evidence for a northern source of deep water in the glacial western North Atlantic. Proceedings of the National Academy of Sciences of the United States of America, 114(11), 2831–2835.
- Menviel, L. C., Spence, P., Skinner, L. C., Tachikawa, K., Friedrich, T., Missiaen, L., & Yu, J. (2020). Enhanced mid-depth southward transport in the Northeast Atlantic at the Last Glacial Maximum despite a weaker AMOC. Paleoceanography and Paleoclimatology, 35(2), e2019PA003793.
- Menviel, L., Yu, J., Joos, F., Mouchet, A., Meissner, K. J., & England, M. H. (2017). Poorly ventilated deep ocean at the Last Glacial Maximum inferred from carbon isotopes: A data-model comparison study. Paleoceanography, 32(1), 2–17.
- Oppo, D. W., Gebbie, G., Huang, K. F., Curry, W. B., Marchitto, T. M., & Pietro, K. R. (2018). Data Constraints on Glacial Atlantic Water Mass Geometry and Properties. Paleoceanography and Paleoclimatology, 33(9), 1013–1034.
- Wilmes, S. B., Schmittner, A., & Green, J. A. M. (2019). Glacial Ice Sheet Extent Effects on Modeled Tidal Mixing and the Global Overturning Circulation. Paleoceanography and Paleoclimatology, 34(8), 1437– 1454.

---

## Author Response (AR2)

**Response to additional reviewer comments: *Simulated stability of the AMOC during the Last Glacial Maximum under realistic boundary conditions* (cp-2020-135)**

We thank reviewer#1 for the additional comments and the particularly helpful suggestions that have helped to further improve the manuscript.

The original reviewer comments are in black and our responses are colored blue. Rephrased or added paragraphs are marked in italics.

Poppelmeier et al. have adequately answered the previous round of comments and have appropriately revised the manuscript. There remain a few ambiguities regarding the novelty of the work that needs to be taken into account in a revised version.

From the title, abstract and discussion, the novelty of the work is not clear and a few statements can be misleading as they would imply that "realistic boundary conditions" have not been used in previous LGM studies: the Bering Strait is closed in most (if not all) LGM simulations. LGM simulations performed with coupled models also include glacial winds (apart from a few which do not include a complex enough atmospheric model). A few LGM studies, particularly model-data comparisons, have also added meltwater into the North Atlantic to weaken the glacial AMOC (similar to the flux adjustment, apart from the fact that no salt was added to the North Pacific, but a debate onto whether there should also be enhanced runoff into the North Pacific during glacial times is out of topic here). The new part in this study is the change in tidal dissipation, which is currently not mentioned in the Abstract, and should be made more obvious throughout. Below are a few suggestions to make sure there is no confusion, particularly for readers who are not numerical modellers. Line numbers refer to the track-change version.

Reply#1: We are grateful to the reviewer for the suggestion to more clearly highlight the novel aspect of considering changes in tidal dissipation for glacial AMOC simulations as well as pointing out statements that were ambiguous. We now revised these aspects throughout the manuscript with particular emphasis on the detailed points listed below.

Title: I would suggest to remove the "under realistic boundary conditions" as this implies that other studies have not been done "under realistic boundary conditions", which is not the case.

Reply#2: We agree with the reviewer on this point and have therefore removed "*under realistic boundary conditions*" from the title.

Abstract: It would be appropriate to mention the tidal dissipation in the abstract as this effect has received little attention.

Reply#3: Indeed, the effect of changed tidal dissipation has rarely been considered for simulations of the LGM AMOC state. We therefore now mention the addition of this process for our LGM simulations also in the abstract as well as in the discussion.

L. 346-348 needs to be amended to make it clear that most LGM simulations already include a closed Bering St and appropriate glacial wind fields. This statement could cause confusion with readers that are not numerical modellers.

Reply#4: We clarified this statement, which now reads: "*The closure of the Bering Strait and wind stress anomalies, are often considered in LGM model runs, andbut changes in tidal dissipation, are neglected in most studies. Our simulations emphasize that all these processes have substantial impact on the ocean circulation here leading to a total increase in AMOC strength of ~5.5 Sv, and a and therefore all of them need to be considered for paleoclimate model simulations.*"

Similarly L. 348-352 should be amended as numerical studies investigating abrupt climate transitions of the last glacial period are performed with a closed Bering Strait, and the impact of Bering St closing is not new.

Reply#5: We agree with the reviewer that investigations of abrupt climate change performed with OGCMs consider the state of the Bering Strait (closed versus open). However, they usually do not investigate transient openings or closings as potential triggers for abrupt climate change. In order to more clearly state this point, we added the following sentence: "*It is thus also important to investigate the impact of transient changes in the state of the Bering Strait, which could act as possible triggers for abrupt climate changes.*"

L. 360-362: This statement needs to be amended as you cannot reach that conclusion in your study. You can say what the Bern3D simulates under the forcing applied, and particularly I would suggest to explicitly state the NADW depth you find, but you cannot say if it is the "true" LGM NADW depth. This is also particularly true as you are not showing/simulating oceanic d13C and comparing it with proxy records. You can also discuss the limitations of your study, since the ocean model of the Bern3D is not an OGCM, and is of coarse resolution both horizontally and vertically.

Reply#6: We feel that in the previous version of the manuscript it was already relatively clear that our inference regarding the LGM AMOC depth might be model specific as we note that "*the simulations presented here suggest that the LGM AMOC did not …*". In order to further clarify this point, we added the following sentence: "*While we note that this conclusion might be model specific, it is in line with revised interpretations of the updated $\delta^{13}C$ data in follow-up studies.*"

L. 362-363: This statement needs to be more precise: What was the previous depth, and what is the new one?

Reply#7: We added the information that the Bern3D model simulates virtually no shoaling between PI and the LGM while nutrient-based proxy reconstructions initially suggested a shoaling of ~1 km.